# Multi-Scale Evaluation of Sleep Quality Based on Motion Signal from Unobtrusive Device

**DOI:** 10.3390/s22145295

**Published:** 2022-07-15

**Authors:** Davide Coluzzi, Giuseppe Baselli, Anna Maria Bianchi, Guillermina Guerrero-Mora, Juha M. Kortelainen, Mirja L. Tenhunen, Martin O. Mendez

**Affiliations:** 1Dipartimento di Elettronica, Informazione e Bioingegneria Politecnico di Milano, 20133 Milano, Italy; annamaria.bianchi@polimi.it (A.M.B.); martin.mendez@uaslp.mx (M.O.M.); 2Unidad Académica Multidisciplinaria Zona Media, Universidad Autónoma de San Luis Potosí, San Luis Potosí 79615, Mexico; guillermina.guerrero@uaslp.mx; 3VTT Technical Research Center of Finland, Tampere, Finland; juha.m.kortelainen@vtt.fi; 4Department of Clinical Neurophysiology, Medical Imaging Centre, Pirkanmaa Hospital District, Tampere, Finland; mirja.tenhunen@pshp.fi; 5Department of Medical Physics, Tampere University Hospital, Medical Imaging Centre, Pirkanmaa Hospital District, Tampere, Finland; 6Laboratorio Nacional—Centro de Investigación, Instrumentación e Imagenología Médica, Facultad de Ciencias, Universidad Autónoma de San Luis Potosí, San Luis Potosí 78210, Mexico

**Keywords:** sleep monitoring, pressure bed sensor (PBS), unobtrusive measure, multi-scale analysis, sleep apnea–hypopnea syndrome (SAHS), shift-working

## Abstract

Sleep disorders are a growing threat nowadays as they are linked to neurological, cardiovascular and metabolic diseases. The gold standard methodology for sleep study is polysomnography (PSG), an intrusive and onerous technique that can disrupt normal routines. In this perspective, m-Health technologies offer an unobtrusive and rapid solution for home monitoring. We developed a multi-scale method based on motion signal extracted from an unobtrusive device to evaluate sleep behavior. Data used in this study were collected during two different acquisition campaigns by using a Pressure Bed Sensor (PBS). The first one was carried out with 22 subjects for sleep problems, and the second one comprises 11 healthy shift workers. All underwent full PSG and PBS recordings. The algorithm consists of extracting sleep quality and fragmentation indexes correlating to clinical metrics. In particular, the method classifies sleep windows of 1-s of the motion signal into: displacement (DI), quiet sleep (QS), disrupted sleep (DS) and absence from the bed (ABS). QS proved to be positively correlated (0.72±0.014) to Sleep Efficiency (SE) and DS/DI positively correlated (0.85±0.007) to the Apnea-Hypopnea Index (AHI). The work proved to be potentially helpful in the early investigation of sleep in the home environment. The minimized intrusiveness of the device together with a low complexity and good performance might provide valuable indications for the home monitoring of sleep disorders and for subjects’ awareness.

## 1. Introduction

Sleep is a biological process intrinsic to life and essential for optimal health as it plays a critical role in brain function and systemic physiology. However, sleep complications and disorders are a growing threat nowadays, affecting up to 70 million people in the United States and approximately 45 million in Europe [1]. Sleep disturbances can involve sleep deprivation and fragmentation [2], occurring when the necessary amount and quality of sleep is not achieved and when there is difficulty in falling asleep [3] or maintaining continuous pattern of sleep [4]. On the other hand, sleep can be affected by other disorder events such as respiratory or motor ones [3].

In this regard, one of the most common and alarming conditions of sleep breathing disorders is Sleep Apnea-Hypopnea Syndrome (SAHS). It affects more adult males with respect to adult females and it is associated with many factors such as overweight and obesity, alcohol, smoking, nasal congestion, and estrogen depletion in menopause, but the only intervention strategy currently supported with enough evidence is weight loss [5,6]. The sleep of subjects suffering of SAHS is characterized by cessations (apnea) or considerable reductions (hypopnea) in respiratory flow. These abnormal episodes are recurrent during the night and can last from a few seconds to minutes [7]. It follows that sleep results are strongly fragmented, whereas other symptoms are excessive sleepiness, decreased cognitive performance, fatigue and also depression [8].

Thus, fragmented sleep can affect the capabilities of memorization, learning and concentration, but also mood and behavior. Due to bad sleep quality, social problems are also frequent such as reduced working efficiency and increased risk in traffic accidents. Importantly, it is also well-known that when the poor sleep condition is prolonged for a long time the risk of developing cardiovascular pathologies such as hypertension increases. For these reasons and the current increase in the number of jobs requiring changing and prolonged shifts, such as nursery, the sleep fragmentation assessment represents a main topic [9,10].

Polysomnography (PSG) is currently the primary method for sleep analysis and is considered the gold standard for sleep monitoring. However, it is an onerous and intrusive technique that can disrupt normal routines. In addition, single nightly measurements of patients, are insufficient to study intrinsic patterns of variability or to correlate sleep with the timing of other activities [11].

With the perspective of minimizing the intrusiveness, m-Health technologies have been developed lately, offering a rapid, customized, and synergistic solution through the use of unobtrusive wearable or home automation devices to monitor vital signs during daily activities [12]. In spite of the fact that great diffusion only occurred in recent years, these devices have found applications in a wide range of scenarios [13] such as fitness or sport [14], rehabilitation [15], health monitoring [16,17] and sleep analysis [18,19] for the aims of prolonged monitoring and preventive interventions.

Different technologies were widely employed for different goals related to sleep analysis such as extracting quality indexes [20], evaluating fragmentation [21] or detecting disorders episodes [22] and sleep phases [23,24]. Methods can be divided according to the devices used, such as electrocardiogram-based [25], actigraphy [26], smartphones [27], smartwatches and complete IMUs [28] or contactless devices, such as bed pressure sensors [7,10,29]. The latter are one of the latest technologies having the advantage of not generating any discomfort. Indeed, these kinds of sensors do not need direct contact with the subject’s body, but they can be integrated into the home environment. Furthermore, the position where the devices are located (smartwatches on the wrists, contactless devices embedded in the bed, near chest or under the mattress) was also evaluated in different studies [24,30,31].

Computational methods used to extract valuable information for screening purposes are mainly based on signal processing and Artificial Intelligence (AI). Common features extracted are averages, ranges, angles, skewness, kurtosis and Wavelet coefficients [32,33], whereas classifiers used are K-Nearest Neighbor (KNN) [34], Decision Tree, Random Forest, Support Vector Machine [10,24,34,35,36] and Hidden Markov Models (HMMs) [37].

In this work, we developed a multi-scale method based on motion signal extracted from an unobtrusive Pressure Bed Sensor (PBS) to evaluate sleep behavior. The contributions of the study are:The implementation of a visualization tool for sleep fragmentation as a function of the activity level;The evaluation of the sleep activity level dynamics from the multi-scale perspective;The sleep quality indexes extracted from the visualization tool and multi-scale analysis which were compared to clinical metrics, such as Sleep Efficiency (SE) and Apnea-Hypopnea Index (AHI);The analysis on motion signal from two different datasets composed of shift-working nurses and people with suspicions of sleep apnea;An easy tool useful for non-invasive devices based on the only motion signal suitable for home monitoring.

## 2. Materials and Methods

### 2.1. Data Acquisition and Study Population

Data used in this study were collected by means of two different acquisition campaigns performed by using the same device, already employed in [7,10,29]. Ethical approval and informed consent details are reported in the cited works.

The PBS device was designed with eight electrodes, located in two columns and four rows, to acquire the measurement of pressure change generated by the sleeping subject. PBS covers a measurement area of 64 cm × 64 cm and it was placed under the mattress at the middle of the sleeping subject’s body. A deepened description of the setup and more details of the device are reported in [7,10,29]. The device was used to acquire:

Dataset 1: includes 22 subjects (11 males and 11 females, age: 48–63 years) that underwent full PSG and PBS recording at the laboratory of the Sleep Centre of Tampere University Hospital (TaUH, Tampere, Finland) for suspected sleep apnea. PSG measured cardiac (ECG), neuronal (EEG), and muscular (EMG) activity. In addition to two elastic bands for Respiratory Inductive Plethysmogram on the thorax and abdomen position, a pulse oximeter for oxygen saturation in blood, thermistor, and nasal cannula for airflow measurement were used during the recording. The Respiratory Event (RE) scoring was performed through an automatic procedure (Rem-Logic software - Embla Systems limited liability company) that detects abnormal events from the nasal airflow signal. For example, apneas are detected as a reduction greater or equal to 90% from the baseline. After the evaluation of the thoracic and abdominal respiratory effort for the classification of the REs, an expert clinician made manual corrections (e.g., false positive/negative REs), if necessary. Each RE present in the recordings was labeled according to four different classes corresponding to the type of RE: (1) Obstructive Sleep Apnea (OSA); (2) Central Apnea; (3) Hypopnea; and (4) Mixed Apnea [7].

Dataset 2: comprises 11 healthy females (age: 20–54 years) that underwent standard PSG and PBS recording at the sleep laboratory of the Finnish Institute of Occupational Health (FIOH, Helsinki, Finland) measuring night or day time sleep for shift workers. Two different recordings, one during daytime sleep after a night shift of work and one during nighttime sleep, were obtained from each subject. The hypnograms of the resulting 22 recordings were then scored by medical specialists following a standard procedure. Each sleep phase was labeled according to the 7 possible classes: (1) Stage 1; (2) Stage 2; (3) Stage 3; (4) Stage 4; (5) REM; (6) Wake with lights off and; (7) Wake with lights on [10].

PBS recording data files gathered were written into a memory card and synchronized with the reference PSG for the analysis. Information about all recordings from both datasets are summarized in Table 1.

### 2.2. Data Conditioning

A signal reflecting the motion and displacement activity occurring during sleep is possible to be captured from the different channels acquired through the PBS.

In Dataset 1, the motion signal was extracted computing the standard deviation for each measurement channel with a sliding raised cosine 4-s window. Then, the average value between channel-wise standard deviations was taken [7]. On the other hand, in Dataset 2, the motion signal was obtained from Principal Component Analysis (PCA) [10]. For both datasets the normalization for the maximum value of the recording was performed.

### 2.3. Pipeline Overview

A multi-scale algorithm using motion signal was designed to assess the sleep quality on the two different datasets. The pipeline can be divided into different steps with the purpose of identifying different states during sleep and analyze their trends at different time scales. After the extraction of the motion signal and the pre-conditioning, the thresholding method is applied to recognize different kinds of activity in various scenarios. Specifically: $TH_{ABS}$ represents the threshold below which subject’s absence from the bed is identified and $TH_{DI}$ is the threshold above which displacements due to subject movements are detected. Afterwards, a multi-scale analysis based on the cumulative histogram of quiet sleep periods is performed to analyze sleep fragmentation to recognize quiet and disrupted sleep. The evaluation is based on prolonged periods of absence of displacements, identified through minQS that represents the minimum duration considered for a quiet sleep interval. A summary of the pipeline is shown in Figure 1.

#### 2.3.1. Motion Detection

The power of the motion signal varies according to the different types of noise that may arise in the environment. Three types of noise identify three situations of interest to be monitored during sleep:External noise: due to the characteristics of the surrounding environment (e.g., traffic). When only this noise is present ($\sigma^{2}<TH_{ABS}$) absence from the bed can be assumed (hereafter called ABS);Physiological noise: due to the natural physiological activity (e.g., breathing) of the subject. If detected ($TH_{ABS}<\sigma^{2}<TH_{DI}$), presence in the bed with no sleep disturbs or movements can be assumed (hereafter called quiet sleep—QS);Displacement: due to physiological movements ($\sigma^{2}>TH_{DI}$) during sleep cycle or abnormal ones (hereafter called DI).Body movements cause the strongest components in the signal, sometimes even saturating the sensor signal, being many orders higher than the other possible components generated by the different noise sources. It is well-known that in typical adult sleep behavior transitions from REM to almost-awake moments generate body movements each 1.5 h that last a few seconds in physiological sleep [38,39]. On the other hand, displacements may also be related to other kind of conditions and scenarios. In particular, the presence of disturbed breathing events (i.e., all thoracic movements stronger than normal physiological activity such as apnea) or abnormal movements (such as myclonias) induce strong fluctuations in the motion signal.The major difference between these cases can be identified through the different duration and periodicity of the events. The abnormal ones are, indeed, more frequent and closer to each other, resulting in shorter periods of disrupted sleep (hereafter called DS). An example of signal highlighting apnea events is shown Figure 2 (box 1).

Therefore, due to the huge differences in the power of the motion signal, the first phase of the algorithm consists of detecting the three main states through the thresholding method. In Figure 2 (box 2), a motion signal showing the differences in power during these distinct states (i) ABS; (ii) QS/DS and (iii) DI and the two thresholds that would identify them is reported. In the figure, it is also highlighted that signal intervals between the two thresholds cannot be considered only related to QS, but also to DS, according to the different duration of periods with no displacements.

#### 2.3.2. Multi-Scale Analysis for Sleep Fragmentation

The only identification of body displacements may suggest potential sleep disorders but in some cases it is fundamental to specifically investigate their characteristics. For this purpose, we introduced the cumulative histogram of QS periods.

The proposed visualization method helps to investigate the duration of these periods, as well as the total amount of QS based on the multi-scale approach. In addition, the disruptions are also easily interpretable and analyzable in their characteristic periodicity. This evaluation of the sleep fragmentation allows to highlight random or specific patterns providing a minimum duration to actually consider a period as QS.

In some cases such as SAHS or myoclonia, threshold-based detection occurs frequently and for a short time. As a consequence, short periods of motion signal below $TH_{DI}$ and between two detected DI events surrounding them (for example, intervals between apnea or abnormal movement events or short stationary periods due to physiological movements) would be correctly detected as DS because of the definition the minimum QS interval. On the other hand, intervals in which the subject is simply lying on the bed would not be considered as QS since they are expected to be characterized by shorter periods of absence of DI. Furthermore, cases in which frequent and long movements occur, not necessarily related to any specific disorder, would be highlighted, identifying a fragmented sleep that may be helpful to be aware of.

Therefore, the exploration of sleep fragmentation through the cumulative histogram of QS periods allows to improve the thresholding-based estimation by accurately identifying real QS ($length(TH_{ABS}<\sigma^{2}<TH_{DI})>minQS$) and DS ($length(TH_{ABS}<\sigma^{2}<TH_{DI})<minQS$). The latter, among all the possible scenarios in which it can occur, is indeed generally related to bad rest periods that it would be crucial to detect and distinguish from QS to correctly monitor the sleep.

In Figure 3, the expected cumulative histograms of QS periods in possible disturbed and healthy good sleep cases are shown. Specifically, it is possible to analyze how much time the subject has spent in periods of QS long at least a certain duration (indicated on the *x*-axis). Reducing this interval, the cumulative duration increases until it is matched to the recording duration. Indeed, the scale of durations is followed by DI and ABS durations which complete 100% of the cumulative. For this reason, the axis is oriented from long to short periods of QS. It is worth noting that the axis starts from periods of 60 min, because an occasional interruption of a longer period does not affect the estimate. It is worth noting that high slope points are marking the step-up of QS interruption below the given duration. This may be a marker of repeated disturbances (e.g., SAHS events or myoclonus) with a period equal or shorter than the step-up point.

The typical sleep pattern is characterized by regular REM/light/deep sleep cycles, thus, it is expected to present no movement other than spontaneous ones occurring during transitions from REM and to result in a modest percentage of fragmented sleep. Conversely, distinct characteristics can be expected and investigated on the cumulative histogram of QS according to the different pathological/disturbed sleep. For example, major percentages of sleep constituted by short periods of QS or significant durations of ABS can be expected in SAHS or insomnia, respectively.

### 2.4. Displacement Analysis and Parameters Optimization

In order to identify the four states of interest (i) ABS; (ii) QS; (iii) DS (iiii) DI, it is necessary to appropriately tune the parameters of the method.

The first parameter to be set is the window size to be considered to evaluate the power of the signal. For this purpose, the distribution of durations of the characterizing DI events were investigated across the two datasets setting different thresholds at 0.01, 0.05, 0.1, 0.2, 0.3, 0.35 and 0.5 on the normalized signal.

From the Probability Density Functions (PDF) shown in Figure 4, it is possible to notice that the durations of the DI extracted from Dataset 1 are distributed up to 20 s. Conversely, in Dataset 2, the majority of the displacement periods (more than 50% of total duration) are segments of 1- and 2-s.

Therefore, the power of the signal was evaluated on 1-s windows as larger intervals would lead to the misdetection of short movements and transitions, which turn out to be very frequent, especially in the Dataset 2.

It is also worth noting that longer DI characterizing Dataset 1 are in agreement with subjects enrolled for sleep problems. Furthermore, the maximum value in the PDF, for almost all thresholds used can be noticed around 5 s, highlighting the typical duration in the order of less than a dozen seconds of the apnea episodes. [6,40]. Conversely, subjects enrolled in Dataset 2 are all healthy resulting in shorter DIs.

Afterwards, the two thresholds and the minimum QS period parameter, described in Section 2.3.1, were set. The optimization was performed using a grid-search based strategy on both datasets. In particular, the best values were found maximizing the correlations between QS and SE and between DS and AHI, when present. In order to have balanced values, both correlation values must be greater or equal than 0.5. Then, the best parameters were obtained maximizing the sum of the two correlations. The resulting values chosen were 0.05 for the threshold to recognize DI and 15 min as the minimum QS period.

### 2.5. Detrended Fluctuation Analysis

A widely-used multi-scale method is the Detrended Fluctuation Analysis (DFA). DFA is a nonstationary time series technique that allows to recognize long-range correlations. It is widely applied in the biomedical field for a variety of applications, such as [41,42]. DFA calculates the root-mean-square fluctuation of time series, disregarding trends and nonstationarities in the data. It allows the detection of intrinsic self-similarity, and it also avoids the spurious detection of apparent self-similarity.

DFA can be divided into three steps. The first one involves the shifting by the mean and the cumulative sum of the time series. The second one consists of dividing it into epochs (scales) of various size (logarithmically spaced) and considering these different segmentations. In the third step, each epoch *e* is detrended and locally fit to a polynomial finding the root mean square $RMS_{e}$, and then the $RMS_{\Delta s}$:(1)$$RMS_{\Delta s}=\sqrt{\frac{1}{N}\sum_{i=1}^{N}{\left[y\left(i\right)-y_{\Delta s}\left(i\right)\right]}^{2}}$$
where *N* is the total number of data points, $RMS_{\Delta s}$ is the root mean square obtained for each scale and *y* is the input signal.

The Hurst exponent *H* is then estimated by computing the linear fit between log-Δs and log-$RMS_{\Delta s}$ as a function of log-*n*. *H* is thus the slope of the line in the range of time scales of interest and can be estimated using linear regression. Through *H* it is possible to quantify the temporal correlations in the signal scale over different window sizes. In particular, whether:*H* = 0.5, the time series is uncorrelated;*H* > 0.5, there are larger fluctuations on longer time-scales than expected by chance, thus long-range correlations;*H* < 0.5, means that fluctuations are smaller in larger time windows than expected by chance, thus the time series is anti-correlated.

Results of DFA applied to the motion signal extracted were compared to indexes from the cumulative histogram of QS periods. For this reason, we logarithmically selected 15 scales from 1 to 60 min, in agreement to the scales considered by our method (see Section 2.3.2).

### 2.6. Experimental Evaluation

Pearson’s correlation analysis was performed between different indexes in multiple scenarios and conditions to assess the extracted sleep quality.

First, the QS extracted index was correlated to *SE*, which was available for all recordings. *SE* is defined as:(2)$$SE=\frac{ST}{TIB}$$
where *ST* is the total sleep time and *TIB* the total time spent “in bed”.

Second, *AHI*, which is the number of apnea and hypopnea events per hour of sleep, was correlated to DS/DI index. *AHI* is defined as:(3)$$AHI=\frac{TNE}{TR}$$
where *TNE* is the total number of apnea and hypopnea events and *TR* is total time duration in hours of the recording. The *AHI* values for adults are categorized as:Normal (N): AHI<5Mild sleep apnea (Mi): 5≤AHI<15Moderate sleep apnea (Mo): 15≤AHI<30Severe sleep apnea (S): AHI≥30

In the correlation analyses, a recording with high values of SE and AHI (Rec.: 3; SE: 0.95; AHI: 40.99) was marked as uncertain. Furthermore, we also considered as uncertain three recordings where hypopneas composed at least 80% of the total abnormal breathing events (see Table 1). These specific abnormal respiratory events, indeed, do not generate any motion [43], thus resulting in being undetectable by PBS.

Furthermore, the two datasets were split according to the SE. Specifically, the threshold was set to 80%, being considered normal/healthy SE above it [44]. It resulted in 19 recordings with good sleep efficiency (GSE), of which 16 are from the Dataset 2 (8 during day and 8 during night) and 3 from Dataset 1 (all with AHI<5) and 21 bad sleep efficiency (BSE), of which 15 are from the Dataset 1 (4 with AHI≥30, 4 with 15≤AHI<30, 2 with 5≤AHI<15 and 5 with AHI<5) and 6 from the Dataset 2 (3 during day and 3 during night).

Correlation analyses were also separately performed on the two datasets to evaluate the presence and the duration of the displacements (DI state). These durations were statistically evaluated through Mann–Whitney tests between the independent subgroups obtained.

In particular, in Dataset 1, the recordings were analyzed together and divided into the normal and mild sleep apnea (N/Mi) group vs. moderate and severe sleep apnea (Mo/S) group. It resulted in groups of 10 N/Mi and 8 Mo/S recordings. In Dataset 2, the recordings were analyzed together and dividing between the ones acquired during the day (11 recordings) and during the night (11 recordings).

Finally, the multi-scale evaluation performed by using the algorithm was compared to DFA. First, the Hurst exponent, that assesses the self-similarity of the time series, was computed and tested through Mann–Whitney tests to assess statistically significant differences across all groups. Then, SE and AHI were correlated to the Hurst exponent.

## 3. Results

First, sleep fragmentation was evaluated through the cumulative histogram of QS periods. In general, this visualization revealed a greater area and a lower slope in subjects with high SE and low AHI. In Figure 5, some example cases together with pie charts of the three main states detected are shown.

It can be noticed that, since the percentage of DI is indicated in the cumulative histogram of QS immediately after the value 1 min on *x*-axis, the recordings having more movements (green area) results in steeper slopes between the last QS period and DI percentage (such as subject 17 reported on middle left). Furthermore, it is worth noting that the area of the cumulative histogram increases together with the percentage of QS (blue area). No differences are visible in the last part of the cumulative histogram of QS periods because the ABS state was never detected since no subject ever stood up during the acquisitions.

All the results obtained by the algorithm for the three main states to be detected recording by recording are then summarized in Table 2.

Afterwards, the Leave-One-Out Cross-Validation was performed on both datasets evaluating the variability in the correlations between SE and QS detected and between AHI and DS/DI detected. Parameters and results found were in line to those obtained through Grid-Search approach described in the Section 2.4. In particular, the resulting correlations between all SE and QS obtained (0.7162±0.0143) and AHI, when available, and DS/DI obtained (0.8537±0.0073) remain stable across all folds.

Afterwards, the agreement between SE and QS and AHI and DS/DI was specifically evaluated through Bland–Altman plots reported in Figure 6 and Figure 7, where uncertain recordings were also marked.

The plots point out differences between each QS and the corresponding SE, and between DS/DI and AHI. In both cases, the mean difference was close to the zero (−0.24 and 0.37, respectively). As regards the comparison between SE and QS, only one recording is out of agreement range (95% range: [−0.70; 0.23]) and it was one of those marked as uncertain. All the other differences resulted in good agreement across a wide range of SE. The other three uncertain recordings were among the recordings that deviated most from the average. Moreover, in the evaluation of AHI in comparison to DS/DI, all differences were within the confidential interval (95% range: [−0.09; 0.82]) with the uncertain recordings among the most deviated ones.

The DI state was then specifically assessed. It is possible to notice that, for the threshold (th=0.05) selected through the procedure described in the Section 2.4, statistically significant differences were found. In particular, the duration of DI differs between Shift-Work and Apnea datasets. Furthermore, this difference was also found in the Apnea Dataset between N/Mi and Mo/S (*p*-value: <0.05) subgroups and in both datasets between GSE and BSE (*p*-value: <0.05) subgroups. A similar duration of displacements resulting in no statistical difference was found in day and night recordings of the Shift-Work Dataset. These results, together with the number of DI for each subgroup, are summarized in Table 3. It is worth noting that the duration of displacements is expressed through the mean and rank, at which outliers corresponding to 5% of the displacements in the least group were removed in the pairwise analysis.

Moreover, the Hurst exponent obtained from DFA was evaluated at dataset- and group-level, as described in the Section 2.6. Larger fluctuations resulting in a higher H value in the Apnea Dataset with respect to the Shift-Work Dataset were found. Similar dynamics were found between groups obtained from the same dataset (N/Mi vs. Mo/S, D vs. N; *p*-value: >0.05), but a statistically significant difference in the self-similarity was found between GSE and BSE groups extracted from both datasets (*p*-value: <0.05). All the results are summarized in Table 4.

Afterwards, the indexes extracted by the algorithm and Hurst exponent were correlated at dataset- and group-level with SE and AHI. All the results obtained from these correlation analyses are summarized in the Table 5.

The correlation between QS and SE resulted to be positive and strong in almost all groups. The Shift-Work Dataset resulted to be greater (0.82) in the case of recordings acquired during the day than those acquired during the night (0.66). In the whole Apnea Dataset, a minor correlation (0.5) with respect to Shift-Work Dataset (0.76) was noticed. At the same time, in this dataset, a strong correlation was found between DS/DI and AHI (0.85), greater in Mo/S (0.68) than N/Mi (0.44). On the contrary, with respect to QS vs. SE, the two subgroup correlations were comparable (N/Mi: 0.53; Mo/S: 0.48). Considering both datasets, QS and SE strongly correlated as previously mentioned (0.72). Conversely, the two subgroups, divided according to the SE show low and comparable correlation values (GSE: 0.4; BSE: 0.39).

As regards as DFA result evaluation, the Hurst exponent was correlated to SE and AHI, when available. In particular, in the Apnea Dataset, H vs. SE found correlation of −0.6 in N/Mi subgroup, −0.41 in Mo/S subgroup and −0.47 in the whole set. AHI was also compared to the results of the DFA but no correlations were found. On the other hand, in the Shift-Work Dataset, only a high positive correlation of 0.75 in the case of night recording was revealed. In general, good negative correlation (−0.53) between H and SE in both datasets was highlighted, resulting in a great difference between GSE (0.34) and BSE (−0.45) subgroups.

## 4. Discussion

In this work, we proposed a multi-scale method to assess the sleep behavior from motion signals acquired through an unobtrusive device. For this purpose, we computed indexes related to the sleep fragmentation at different temporal scales and evaluated them through the comparison to clinical indexes. The complexity of the method is low, the hardware requirements are low-cost and the four indexes of quality estimated are easily interpretable and informative for users in everyday life.

The multi-scale analysis provided a visualization of sleep fragmentation and a tool to identify states of interest during sleep, with particular attention to the definition of quiet/disrupted sleep (QS/DS). In fact, although numerous valuable indexes are often estimated through objective measures from different devices [20,21,22,23,24], the recognition of real periods of QS is fundamental and may represent an easily interpretable indication for the subject, especially in home monitoring. In different sleep pathologies, multi-scale components of sleep fragmentation are difficult to be recognized and more informative visualizations would be essential in clinics to better interpret the pathology of a specific patient and its characteristics. For example, in the case of a Chronic Obstructive Pulmonary Disease subject, how long is the interval between two apneas? The cumulative histogram of QS periods extracted from the motion signal allowed us to analyze the sleep patterns in comparison to healthy subjects and to visualize differences in sleep fragmentation.

### 4.1. Sleep Quality Indexes Assessment

From the different examples in Figure 5 some important characteristics were enhanced. As expected, a minimal DI percentage was found in healthy subjects due only to the spontaneous movements before REM phases (brief awakenings), characterizing a typical no-disturbed sleep phase pattern [38,39]. Other general relevant properties observed in these cases were that 50% of sleep is composed of QS periods of more than 30 min and that the movements are exclusively composed of physiological ones that fragment in short periods of QS a modest percentage of sleep. Conversely, during different kinds of disturbed sleep it can be noticed that:Total time spent in DI state is greater than in the case of healthy sleep;Long periods of QS with an absence of DI constitute a small percentage of the night and fragment a modest percentage of sleep into short periods of QS;The point of maximum slope characterizes the dynamics of fragmented sleep.

In particular, the latter represents the minimum QS period to be considered to assure that real QS and DS periods are identified. In the datasets analyzed, 15 min was found to be the best value to distinguish healthy and pathological sleep. However, it is worth noting that, for the pathological cases, this point can significantly change according to the different nature and severity of the disturbance. For example, the maximum slope in the reported examples in Figure 5, although being less than 15 min, varies from being very close to the 0 min in the figure on the middle left (subject 17), to almost 15 min on the top left (subject 14). This value corresponds to the fastest change in the cumulative histogram, thus it is expected to underline the most frequent and characteristic time interval fragmenting the QS of the subject. The present insight points out the necessity of re-calibrating this parameter for the specific sleep disorder to enable an optimal recognition. At the same time, this result confirmed:The validity of the cumulative histogram of QS periods as a tool for the qualitative investigation of sleep fragmentation during a night of sleep;Its worthiness in longitudinal studies, whatever the chosen period is. In fact, although different sleep disorders can have different and specific dynamics, it is possible to highlight quality trends, showing improvements and worsenings among multiple days.

Furthermore, these findings on the cumulative histogram of QS periods demonstrated that a multi-scale analysis is needed when analyzing sleep from motion signal.

Indeed, for the purpose of assessing QS and DS, we dealt with the problem of the definition of a gold standard. These represent particular states during sleep which are easily interpretable but that, to the best of our knowledge, were not explored through objective measures from contactless devices. In this regard, what is a real period of quiet sleep is not straightforward since in many sleep disorders there are intervals of stillness that may be not actually quiet, as above-mentioned for apparent quiet sleep between abnormal breathing/movement events [40]. Another concern about this finding is about the intrinsic limitation of motion signal. It is indeed impossible to distinguish between real QS periods and intervals in which the patient is completely still but awake. This is an intrinsic limitation of the technology [24] but the identification of a minimum QS period can also improve the robustness of the methods in these cases. On one hand, by correctly setting this value, real QS periods are detected by verifying that they are enough long to be considered undisrupted. On the other hand, it is unlikely that an awake person remains totally still and with completely regular breathing for more than 15 min. Either way, the awareness on this definition of QS must be considered.

To have a direct relation of QS and DS with gold-standard, qualitative and quantitative analyses were carried out to show the agreement. From a first visual exploration of the results (in Table 2) it can be noticed that QS in agreement to SE, shown in Table 1, is higher in Dataset 2 than Dataset 1. On the other hand, the DI state appears to be much less present in Dataset 2, which is consistent with known characteristics of motion signal in SAHS [7]. Indeed, subjects from Dataset 1 were acquired for sleep problems, resulting in numerous members of the group suffering from SAHS and, thus, several abnormal movements. Furthermore, a higher DS/DI tends to be associated with a higher AHI and a reduced SE. A clear example is given by the comparison of recordings 6 and 19. Second, the correlations between SE and QS and DS/DI and AHI of all recordings resulted to be high and with low variability in cross-validation. This result was also confirmed by the Bland–Altman Plot in Figure 6 and Figure 7, where all recordings resulted to be in the agreement range, except for one case, also marked as uncertain. In general, all uncertain recordings were among the most deviated ones. This may suggest a good correlation with the proposed measures, unless unexpected scenarios of SE and AHI and the intrinsic limitation of hypopneas recognition. It is indeed well known that this kind of event can be difficult to be detected by different devices and technologies [45,46], and, especially in motion signals where differences cannot be visualized [43].

The motion signal, indeed, reflects the activity occurring during sleep, capturing all kinds of movement, proving to correlate to wake stage periods [30,47]. The presence of movement was thus tested on the datasets available through DI state to point out possible valuable characteristics of the subgroups. In particular, it can be noticed that in Table 3 differences were found between all subgroups considered. In particular, in the case of splitting through AHI and SE the differences were found to be statistically significant, while dividing by timetable (Day vs. Night) not. Although a slight difference in DI durations resulting in a bit higher variability during day was found (mean ± std; D: 2.19 ± 1.47; N: 2.17 ± 1.35), the number of DI events per hour in the whole of Dataset 2 was higher during the night (D: 10.42; N: 13.84). This seems to confirm the similar results between daytime and nighttime sleep as in [9,10], especially in subjects adapted to the shift-works. In a future perspective, the algorithm may be employed in long-term monitoring at home according to the different shifts and to assess the adaptation to these. Unobtrusive technologies may be of unvaluable interest for the prevention of the well-known risks of occurrence of coronary heart and cardiovascular disease, and beyond that, psychomotor and mood problems [9,10,48].

### 4.2. Multi-Scale Analyses Comparison

Afterwards, the DFA multi-scale method was applied to investigate differences in the Hurst Exponent. First, it is worth noting that no significant changes were found in Table 4 between subgroups of the same datasets. D and N recordings show similar dynamics in agreement to similar SE values in the two groups (D: 0.83 ± 0.14; N: 0.83 ± 0.09) but also duration and number of DI per hour, as above mentioned. H also did not discriminate N/Mi and Mo/S apnea patients. It is worth noting that 7 of the 10 recordings within N/Mi had healthy AHI, but only 3 of these had healthy SE (≥80%). This result may suggest a reduced sleep quality due to possible other reasons [49], although a low number of apnea episodes occurred. This bias in the results seems to be also confirmed by the analysis performed dividing both datasets between high and low SE. These two subgroups were composed of 19 and 21 subjects, respectively, where the first included the 3 subjects from Dataset 1 considered healthy according to SE and 16 subjects from Dataset 2. This may suggest that self-similarity significantly grows in fragmented sleep, presenting larger fluctuations.

Table 5 showed the correlations found between clinical indexes and computed ones at the group- and subgroup-level. For example, for QS vs. SE a better correlation of daytime recordings was noted, which may be associated to a slightly less variable SE (SE-D: 0.83±0.14; SE-N: 0.83±0.08) due to shorter recordings. For other cases, slight general greater correlation was found in subjects that slept better (N/Mi and GSE), which is probably associated with motion signals from Mo/S and BSE being more variable, in general. For this reason, in cases of bad sleep it is easier to correctly recognize DS/DI, also mirroring the better correlation with AHI for Mo/S subgroup. Furthermore, bad sleep, in general, can be caused by a number of reasons [49]. For example, although cases of recordings with high percentage of hypopneas were excluded, in remaining ones they can still be present and produce false QS periods. To deeply investigate the hypopneas, the abnormal breathing events in the two subgroups of N/Mi and Mo/S were analyzed. Although in Mo/S the number and the duration was clearly greater, the percentage of hypopneas with respect to total duration of abnormal breathing events was less than N/Mi. In particular, 0.39% of the total duration of all breathing events in Mo/S were hypopneas, whereas 0.58% of the duration in N/Mi were hypopneas, hence resulting in a more difficult recognition of DS and higher QS periods identification.

Another interesting finding was that the Hurst exponent resulted to negatively correlate to SE (−0.53). In agreement with previous result, *H* appears to grow as SE decreases, and it is worth noting that this value is very similar in Mo/S and BSE subgroups (Mo/S: −0.41; BSE: −0.45). The latter indeed contains all recordings of Mo/S (8/21) but also seven from N/MI and six from Dataset 2. These cases appear to not heavily affect the result obtained in Mo/S group and consistency among all unhealthy subjects. In general, this may suggest an auto-affine structure in motion signal of SAHS cases, given by the known periodical pattern [50,51], which is not present instead in low SE cases. On the other hand, a positive good correlation was found between H and SE during night recordings of Shift-Work Dataset (0.75). In D group and whole Dataset 2 this correlation was not found. It is worth noting that the N group comprised the longest acquisitions and with good and the least variable SE across all subjects, resulting to be the most homogeneous group. In this case, a greater agreement between H and SE is suggested and it can be speculated to mirror the less clear auto-affine structure in shorter recordings. Furthermore, we may also speculate that H appears to be prone to great changes according to variable SE values. Furthermore, this could possibly explain the great difference between the values in GSE and BSE, since GSE is mainly composed by these recordings from Dataset 2, joined with three recordings from Dataset 1 containing few apnea episodes. On the other hand, BSE is mainly composed of recordings from Dataset 1, resulting in huge differences in correlations obtained in the use of DFA, probably due to such different apnea case patterns.

### 4.3. Home Monitoring Perspectives

The results cast a new light on the home sleep assessment measures obtained from unobtrusive devices that may be intuitively monitored by the subject. Recently many studies focused on the problem of minimizing intrusiveness [52], especially during sleep [53]. The problems of intrusiveness and conditioning related to PSG are well-known [54,55], thus the continuous screening through home devices results to be fundamental, especially considering the latest development of these technologies [56]. Moreover, PBS has the advantage of eliminating this problem. Numerous devices for sleep monitoring were successfully developed in recent years, such as smartwatches and waist or chest belts [20,24,27], but with the discomfort of wearing them during the whole sleep night. Although, it may be considered only a small limitation, its continuous use in daily living is discouraged. Conversely, contactless devices do not need direct contact to the patient’s body, not generating any discomfort and reaching good performance.

In Table 6, some state-of-the-art studies are reported to compare the proposed work in terms of technologies, methods, datasets used, detected indexes and advantages. In particular, the studies were selected according to the datasets used and to characterize the most widespread and valuable sleep indexes extracted in literature. It is worth noting from the table that the studies based on EEG, ECG and PPG signals [32,36,57,58,59,60,61,62] can be used to extract valuable information on sleep stages or sleep apnea; however, they need higher computational cost and specialized devices for signal acquisition. On the other hand, other works based on motion signal from accelerometer and PBS [7,10,24,63,64] underline the advantage of causing low or mild discomfort to generally detect sleep and wake phases. However, the proposed work based on PBS allows us to characterize the sleep activity level dynamics from the multi-scale perspective and to provide interpretable indexes for the continuous home monitoring, based only on the motion signal.

It is worth noting that home assessment through these devices must be employed carefully. PSG is the gold-standard for sleep analysis and m-Health technologies may be helpful in raising a first alarm. Indeed, subjects suffering from many sleep disturbances are not often aware of their condition resulting in fatigue, low concentration and memorization [65]. In other cases, there is hope for the clinicians that biomarkers and other indicators will help diagnose presymptomatic signal of diseases. It was indeed found for example that Parkinson’s Disease can be associated with Restless Leg Syndrome [66,67,68]. It follows that its preventive identification would be of great importance.

Furthermore, as above mentioned, the algorithm results to be particularly helpful for longitudinal study and, in general, to have an easy monitoring of personal sleep. It could be helpful, for example, to visualize the sleep fragmentation of specific disturbed nights or analyze the trend of QS/DS in correspondence to the introduction of preventive measures. Examples may be the better care of personal sleep hygiene, such as making sport [69,70], avoiding the use of electronic devices before sleeping [71], or in the worst scenarios, the introduction of sleeping medication.

### 4.4. Accelerometer Experimentation and Adaptability

The present study was conceived with the aim of also identifying periods of absence from the bed, which may be particularly helpful in cases of subjects with insomnia disorder or that awakens multiple times during night [72]. Due to the fact that no subject in either dataset ever got out of bed during the recordings, this investigation was not possible on the presented data. This points out the importance of home monitoring, since the acquisition conditions in a controlled environment do not perfectly mirror the real conditions. This state was qualitatively assessed through an experimentation performed on a prototypal device with a triaxial accelerometer, designed to monitor the sleep of subjects during daily living. In particular, the recognition of the state of absence from the bed was achieved through a manual setup, lasting 1 min. The device acquired data for 1 min with and without the subject on the bed and setting the threshold through a ROC analysis. Due to the significant difference between the only external noise, due to traffic and environmental conditions, visible when subject is not lying on the bed, and physiological noise, due to breathing for example, the identification did not result in relevant errors. All the results on the other states representing sleep indexes resulted to be in line with PBS performance, confirming the adaptability of accelerometer data.

## 5. Conclusions

In this work, we studied the multi-scale behavior of the motion signal extracted from PBS during sleep. The experimentation conducted on two different datasets acquired from shift-working nurses and people with suspicions of sleep apnea was assessed in correlation to clinical indexes and compared to a multi-scale method. The entire pipeline is suitable for online computation on an unobtrusive device dedicated to the described purpose of avoiding any discomfort to the subject. This may provide valuable indications in daily living for a rapid and continuous screening of sleep through a home device.

## Figures and Tables

**Figure 1 sensors-22-05295-f001:**
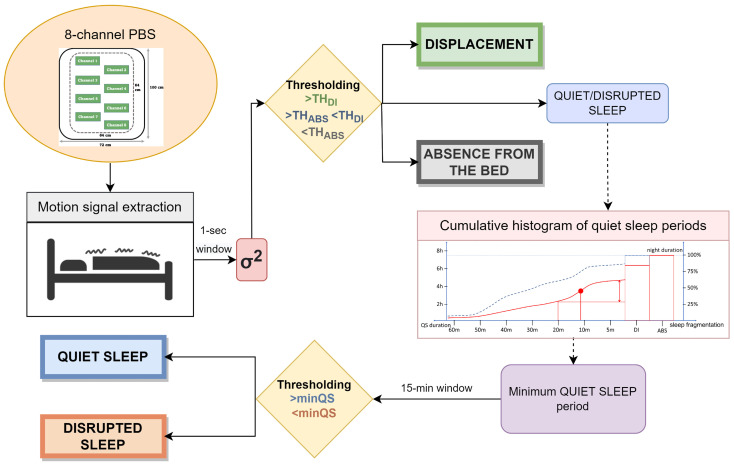
Complete pipeline of the designed algorithm.

**Figure 2 sensors-22-05295-f002:**
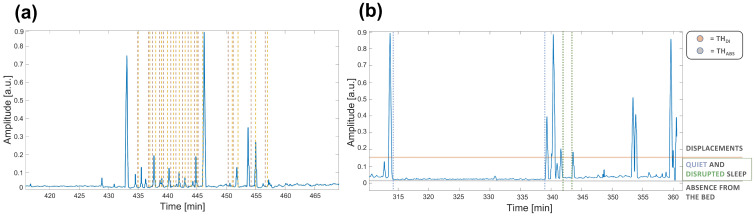
Example of motion signals on time intervals of about one hour of the rec. 2. In (**a**) the labeled apnea events are shown. Brown dashed lines represent the event starting, while yellow dashed lines the ending. In (**b**) the two thresholds are shown to highlight the different sources of noise. In particular, $TH_{DI}$ (horizontal line in red) is the threshold above considering displacements, whereas $TH_{ABS}$ (horizontal line in gray) is the threshold below considering absence from the bed because of the reduced activity due to the only external noise. The activity between the two thresholds highlights the period spent lying on the bed that can identify QS and DS. Furthermore, a long time interval identifying QS is highlighted between the two dashed blue vertical lines, while a short time interval identifying DS is shown between the two dashed green vertical lines.

**Figure 3 sensors-22-05295-f003:**
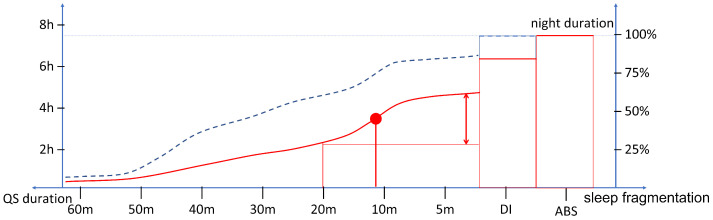
Schema representing the possible cumulative histogram of QS periods in disturbed (red) and healthy good (dashed blue) sleep. The point of maximum slope (red dot) is expected to characterize the dynamics of the fragmented sleep.

**Figure 4 sensors-22-05295-f004:**
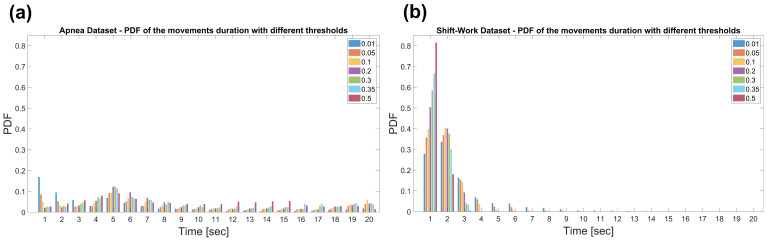
Probability Density Function (PDF) of the displacements duration in the Apnea Dataset (**a**) and the Shift-Work Dataset (**b**). The durations were obtained setting thresholds at 0.01, 0.05, 0.1, 0.2, 0.3, 0.35 and 0.5.

**Figure 5 sensors-22-05295-f005:**
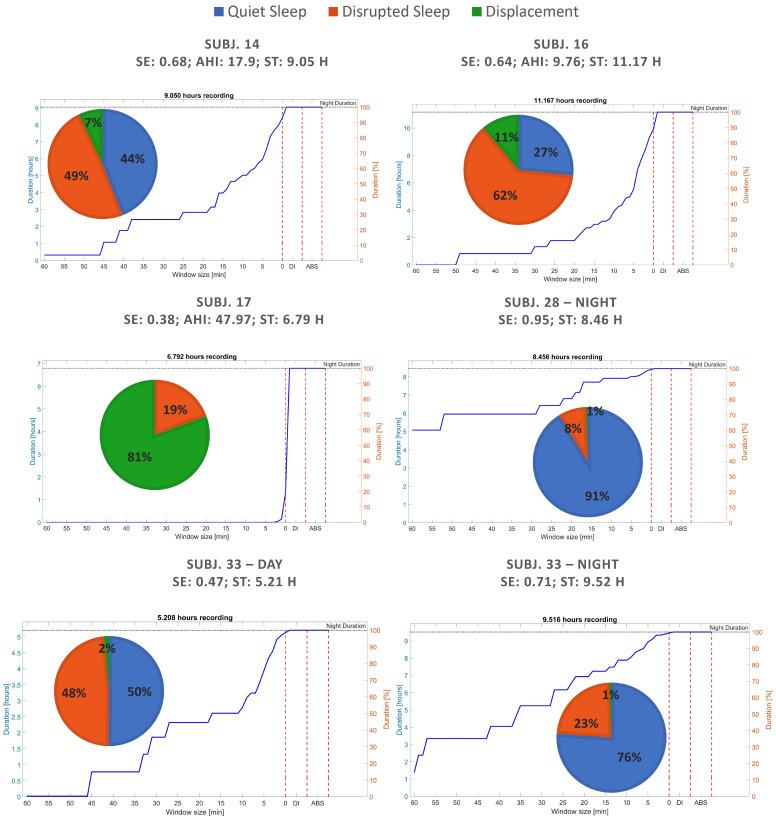
Sleep quality evaluation and fragmentation of recordings from both datasets through pie charts and cumulative histogram of QS periods.

**Figure 6 sensors-22-05295-f006:**
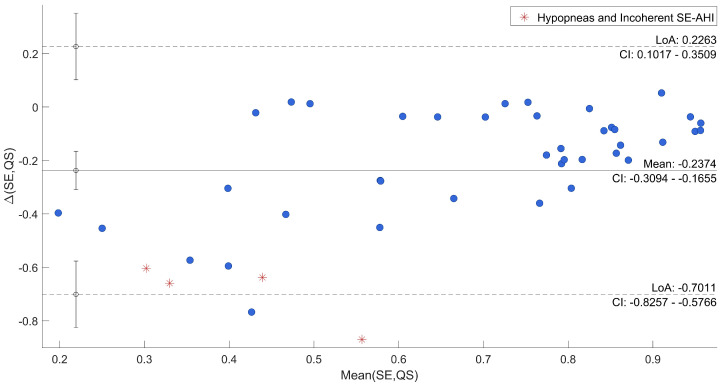
Bland–Altman Plot of SE vs. QS.

**Figure 7 sensors-22-05295-f007:**
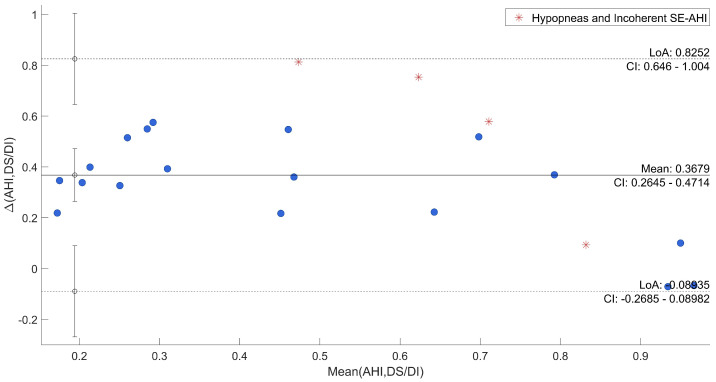
Bland–Altman Plot of AHI vs. DS/DI.

**Table 1 sensors-22-05295-t001:** Characteristics of the datasets.

1. Apnea Dataset						2. Shift-Work Dataset				
**Rec.**	**Subj.**	**ST (h)**	**SE**	**TNE**	**AHI**	**Rec.**	**Subj.**	**ST (h)**	**SE**	**Timetable**
1 *	S1	6.01	0.72	21	3.49	23	S23	4.34	0.95	D
2	S2	9.66	0.77	145	15.01	24	S23	9.00	0.83	N
3 *	S3	8.98	0.95	368	40.99	25	S24	3.90	0.84	D
4	S4	8.74	0.42	2	0.23	26	S24	9.83	0.85	N
5	S5	7.64	0.44	1	0.13	27	S25	4.94	0.85	D
6	S6	8.87	0.66	454	50.63	28	S25	8.36	0.69	N
7	S7	7.22	0.63	13	1.80	29	S26	4.06	0.69	D
8	S8	8.34	0.59	6	0.72	30	S26	8.37	0.89	N
9	S9	9.65	0.68	5	0.52	31	S27	4.89	0.86	D
10	S10	6.18	0.46	196	31.74	32	S27	9.05	0.83	N
11	S11	6.61	0.61	345	52.21	33	S28	5.54	0.94	D
12	S12	6.49	0.53	180	27.75	34	S28	8.46	0.95	N
13 *	S13	7.69	0.58	99	12.87	35	S29	5.25	0.93	D
14	S14	9.05	0.68	162	17.90	36	S29	8.68	0.75	N
15 *	S15	7.32	0.63	161	22.00	37	S30	4.13	0.93	D
16	S16	11.17	0.64	109	9.76	38	S30	8.09	0.90	N
17	S17	6.79	0.38	319	46.97	39	S31	4.60	0.86	D
18	S18	8.56	0.90	39	4.56	40	S31	9.23	0.85	N
19	S19	8.18	0.87	27	3.30	41	S32	4.80	0.79	D
20	S20	7.02	0.77	161	22.92	42	S32	7.86	0.92	N
21	S21	8.40	0.91	1	0.12	43	S33	5.21	0.47	D
22	S22	5.73	0.80	34	5.93	44	S33	9.52	0.71	N

ST: Sleep Time in hours; SE: Sleep Efficiency; TNE: Total Number of Events; AHI: Apnea-Hypopnea Index; The recordings marked with “*” symbol are the recordings considered uncertain (see the Section 2.6 for the selection of the uncertain recordings).

**Table 2 sensors-22-05295-t002:** Sleep quality indexes detected by the proposed algorithm for each recording of the two datasets.

1. Apnea Dataset					2. Shift-Work Dataset				
**Rec.**	**Subj.**	**QS (%)**	**DS (%)**	**DI (%)**	**Rec.**	**Subj.**	**QS (%)**	**DS (%)**	**DI (%)**
1 *	S1	12.02	51.46	36.52	23	S23	90.42	8.92	0.66
2	S2	35.23	49.81	14.96	24	S23	71.37	27.34	1.29
3 *	S3	12.14	81.02	6.84	25	S24	93.64	5.43	0.93
4	S4	42.05	47.10	10.85	26	S24	79.78	19.66	0.56
5	S5	48.26	41.38	10.36	27	S25	81.31	17.99	0.70
6	S6	10.15	72.20	17.65	28	S25	68.34	30.61	1.05
7	S7	62.73	32.11	5.16	29	S26	73.18	25.99	0.82
8	S8	58.72	36.49	4.79	30	S26	79.03	20.08	0.89
9	S9	44.06	45.47	10.47	31	S27	81.28	18.13	0.59
10	S10	2.30	84.54	13.16	32	S27	68.44	30.76	0.80
11	S11	6.69	7.23	86.08	33	S28	92.63	6.94	0.43
12	S12	24.62	56.64	18.74	34	S28	91.23	8.34	0.43
13 *	S13	0.02	82.63	17.35	35	S29	77.15	22.40	0.45
14	S14	44.02	48.89	7.09	36	S29	74.65	24.63	0.72
15 *	S15	0	60.32	39.68	37	S30	84.57	14.87	0.56
16	S16	26.58	62.37	11.05	38	S30	77.02	22.19	0.79
17	S17	0	19.48	80.52	39	S31	68.61	30.76	0.63
18	S18	58.62	37.71	3.67	40	S31	69.66	29.31	1.03
19	S19	71.82	24.86	3.32	41	S32	82.21	17.14	0.65
20	S20	4.26	70.11	25.63	42	S32	92.58	6.87	0.55
21	S21	65.16	30.89	3.95	43	S33	50.16	48.39	1.45
22	S22	49.38	45.14	5.48	44	S33	76.13	23.27	0.60

The recordings marked with “*” symbol are the recordings considered uncertain (see the Section 2.6 for the selection of the uncertain recordings).

**Table 3 sensors-22-05295-t003:** Displacements extracted.

Displacements				
1. Apnea Dataset				
Dur	N/Mi (n = 10)	Mo/S (n = 8)	*p*	Wh
mean [rank] (s)	22.93 [1, 63]	26.85 [1, 110]	<0.05	25.54 [1, 110]
*n*. DI	1494	2973		4467
2. Shift-Work Dataset				
Dur	D (n = 11)	N (n = 11)	*p*	Wh
mean [rank] (s)	2.19 [1, 6]	2.17 [1, 5]	>0.05	2.17 [1, 6]
*n*. DI	607	1271		1878
Both				
Dur	GSE (n = 19)	BSE (n = 21)	*p*	Wh
mean [rank] (s)	4.65 [1, 21]	24.12 [1, 111]	<0.05	18.62 [1, 111]
*n*. DI	1791	4554		6345

Dur: duration in seconds; n. DI: number of displacements; Wh: whole dataset; n: number of recordings. Non parametric (Mann–Whitney test). In GSE: 16 are from Dataset 2—8 D and 8 N—and 3 from Dataset 1—all N. In BSE: 15 are from the Dataset 1 — 4 S 4 Mo, 2 Mi and 5 N — and 6 from the Dataset 2 — 3 D and 3 N.

**Table 4 sensors-22-05295-t004:** Self-similarity through Hurst Exponent (H) computation.

Hurst Exponent				
1. Apnea Dataset				
H	N/Mi (n = 10)	Mo/S (n = 8)	*p*	Wh
mean ± std	0.76±0.07	0.75±0.05	>0.05	0.75±0.06
2. Shift-Work Dataset				
H	D (n = 11)	N (n = 11)	*p*	Wh
mean ± std	0.63±0.06	0.65±0.04	>0.05	0.64±0.05
Both				
H	GSE (n = 19)	BSE (n = 21) ${}^{6}$	*p*	Wh
mean ± std	0.65±0.05	0.73±0.08	<0.05	0.69±0.08

Wh: whole dataset; n: number of recordings. Non parametric (Mann–Whitney test). In GSE: 16 are from Dataset 2 — 8 D and 8 N — and 3 from Dataset 1 — all N. In BSE: 15 are from the Dataset 1 — 4 S 4 Mo, 2 Mi and 5 N — and 6 from the Dataset 2 — 3 D and 3 N.

**Table 5 sensors-22-05295-t005:** Correlation analyses.

Correlation Analyses									
	**1. Apnea Dataset**			**2. Shift-Work Dataset**			**Both**		
	**N/Mi (n = 10)**	**Mo/S (n = 8)**	**Whole**	**D (n = 11)**	**N (n = 11)**	**Whole**	**GSE (n = 19)**	**BSE (n = 21)**	**Whole**
QS-SE	**0.53**	0.48	**0.50**	**0.82**	**0.66**	**0.76**	0.40	0.39	**0.72**
DS/DI-AHI	0.44	**0.68**	**0.85**	na	na	na	na	na	na
H-SE	**−0.60**	−0.41	−0.47	−0.17	**0.75**	0.07	0.34	−0.45	**−0.53**
H-AHI	−0.21	0.30	−0.02	na	na	na	na	na	na

Pearson’s correlation (good correlation in bold). na: unavailable results because of missing AHI; n: number of recordings. In GSE: 16 are from Dataset 2 — 8 D and 8 N — and 3 from Dataset 1 — all N. In BSE: 15 are from the Dataset 1 — 4 S 4 Mo, 2 Mi and 5 N — and 6 from the Dataset 2 — 3 D and 3 N.

**Table 6 sensors-22-05295-t006:** State of the art comparison.

State of the Art						
**Reference**	**Year**	**Device**	**Method**	**Dataset (n. sub)**	**Detected Indexes**	**Advantages**
Proposed work	2022	PBS	Multi-Scale Signal Processing based method	33 (HC vs. SAHS vs. SW)	ABS, QS, DS, DI	No discomfort, interpretability, model complexity
Hussain et al. [57]	2022	EEG	MLP	154	Sleep stages	Performance, low number of channels, no feature extraction
Yang et al. [58]	2022	ECG	1D-SEResGNet	25 (HC vs. SAHS)	OSA	Embeddable in wearable, no feature extraction
Wu et al. [59]	2021	PPG (wrist)	IBS for fluctuation analysis, RFC	92 (HC vs. SAHS)	AHI	Mild discomfort, interpretability
Banfi et al. [63]	2021	ACC (wrist)	CNN	81	Sleep vs. Wake	Mild discomfort, no feature extraction
Baty et al. [36]	2020	ECG belt	SVM	241 (HC vs. SAHS)	AHI	Mild discomfort, interpretability
Hulsegge et al. [64]	2019	2 ACC (thigh, ankle)	LMM and GEE logistic regression	194 (SW vs. non-SW)	Onset, Offset, TST	Mild discomfort, interpretability, model complexity
Mendez et al. [10]	2017	PBS	SVM	6 SW	Sleep Stages	No discomfort, interpretability, model complexity
Aktaruzzaman et al. [24]	2017	ACC (wrist), HRV	SVM	18 HC	Sleep vs. Wake	Mild discomfort, interpretability, model complexity
Mora et al. [7]	2015	PBS	Signal Processing based method	24 (HC vs. SAHS)	AHI	No discomfort, interpretability, model complexity

EEG: Electroencephalography; ECG: Electrocardiography; PPG: Photoplethysmography; ACC: triaxial accelerometer; HRV: Heart Rate Variability; MLP: Multilayer Perceptron; 1D-SEResGNet: one-dimensional squeeze-andexcitation residual group network; IBS: Information-Based Similarity; RFC: Random Forest Classifier; CNN: Convolutional Neural Network; SVM: Support Vector Machine; LMM: Linear Mixed Models; GEE: Generalized estimation equations; SW: Shift Workers.

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
