# Peer review of "Multi-Scale Evaluation of Sleep Quality Based on Motion Signal from Unobtrusive Device"

_sensors, 2022, doi:10.3390/s22145295_

Round 1
Reviewer 1 Report
This study aimed to utilize an unobtrusive pressure bed sensor to evaluate sleep postures and sleep quality. I have the following suggestions.
- What is the novelty of this study although several pressure sensor-based studies have been proposed earlier for sleep status recognition? Please write down the contribution of the study at the end part of the Introduction section in bulleted form.
- Authors should describe recent studies related to sleep stage prediction. For example, EEG is investigated for Sleep stage prediction in article, quantitative evaluation of eeg-biomarkers for prediction of sleep stages.
- Authors should improve the conceptual figure of data processing with more details and model parametrization.
- Figure quality needs to improve.
- Authors should introduce the biosignal applications in sleep monitoring for stroke prediction in the article, healthsos: real-time health monitoring system for stroke prognostics.
- Authors should describe signal processing and threshold detection approaches in an appropriate mathematical format.
- Authors must make discussion on the advantages and drawbacks of their proposed system with other studies adding a comparative table in discussion section.
Reviewer 2 Report
The paper is presenting a new type for data analysis based on multi-scale evaluation of motion signals taken with a pressure bed sensor. Comparing the data to a "gold standard" obtained by polysomnography measures for sleep quality are derived. To different groups of subjects are investigated with this sensor and the data evaluation methods. The comparison shows that this unobstrusive method provides reasonable information about the sleep quality and thus can be used for unsupervised characterization of sleep quality.
The methods are well explained, only minor issues needs some improvements and explanations. The results and methods are explained in 7 figures and 5 tables.
The minor improvements are recommended. For details see enclosed pdf where questions and suggestions are provided as comments. Additionally, the highlighted text should be checked.
Specifically the authors are suggested to:
- to change in line 88 reference [29] to [7] as the PBS is explained in [7], not in [29]
- line 100: what kind of corrections were these, essential to analysis?
-related to line 144/145 (hint to saturation): how critical is saturation for the data evaluation? Due to normalization to max. and threshold setting related to normalized values, saturation might cause a somewhat "random" threshold" setting. How is this avoided (is saturation recognized?)
- Fig. 2: unit of amplitude (suppose a.u. or normalized to max)
-Fig. 3: are in Fig. 3 schematic plots shown or real data? In case of real data it is strange that disturbed and healthy good sleep show similar behavior ("steps" at same QS duration, parallel curves). Additionnally, the definition for the values of DI and ABS are not clear: in the following plots both DI and ABS are at 100% level (what does this mean?`)
-Fig 5: font size for axes title and number much to small. Why are Di and ABS always 100%?
- Fig. 6 and 7: provide scale of y-axis (e.g. -0,711 and 0.2263 at according dashed lines on the y-axis
- table 3 needs more explanation: for the given std-values the mean values sometimes seem to be not reliable! Should be at least commented.
line 397/398: what does "slope of 15 min" mean?
-line 418: the situation of a subject lying awake in the bed should be recognized by other unobstrusive sensors e.g. for BCG (heart rate and respiration). Did the authors check for commercial (e.g. Beuerer sleep expert) as well other published work on heart rate and respiration rate measuerement? This might provide further helpful information!
-line 540/541: what does this mean? No data/no event where subject was out of bed (ABS) or lack of signal?

Reviewer 3 Report
(1)How does sleep apnea information cause changes in signals of Pressure Bed Sensor(PBS)? An example diagram needs to be added to show the PBS signal difference between apnea and normal breathing.
(2) Compared with ECG or PPG signal, what are the advantages and disadvantages of PBS signal for AHI measurement.
(3)...DS/DI positively correlated (0.85±0.007) to Apnea-Hypopnea Index (AHI).... In the discussion, it is suggested to compare the correlation between a single feature of different signals (e.g. ECG, PPG) and AHI correlation, including wearability / number of signal channels / signal quality / highlights. Further refine the optimization of DS/DI measurement and the advantages of PBS signal. Discuss method motivation and refine the innovation.
e.g.
ECG signal from PSG: l Teporal dependecy complexity: The Novel Approach of Temporal Dependency Complexity Analysis of Heart Rate Variability in Obstructive Sleep Apnea, Computers in biology and medicine 2021...
wearable PPG/ECG: Sleep apnea screening based on Photoplethysmography data from wearable bracelets using an information-based similarity approach. Computer Methods and Programs in Biomedicine 2021
Classifification of Sleep Apnea Severity by Electrocardiogram Monitoring Using a Novel Wearable Device, Sensors 2020
(4) At the same time, the machine learning method is widely used in AHI detection, and has achieved significantly better results. The advantages and disadvantages of the paper'method need to be analyzed and discussed.
For example:l Obstructive sleep apnea detection from single-lead electrocardiogram signals using one-dimensional squeeze-and-excitation residual group network. Computers in biology and medicine 2022(CNN)
A dual-model deep learning method for sleep apnea detection based on representation learning and temporal dependence. Neurocomputing 2022.(CNN+GRU)
Obstructive sleep apnea detection from single-lead electrocardiogram signals using one-dimensional squeeze-and-excitation residual group network. Computers in biology and medicine 2022(CNN)
Reviewer 4 Report
The authors describe the detail signal process method (with optimization threshold) for the multi-scale evaluation of the sleep quality based on motion signal measured by a pressure bed sensor (PBS). The paper can be published in Sensors after minor revision. Here are some comments:
The author concludes that the quiet sleep index shows 0.72 correlation with sleep efficiency index, what does the 0.72 correlation means to the reader? Is it high enough to use the PBS to replace PSG in some applications? What are those applicaitons?
This is similar to the 0.85 correlation between DS/DI index and Apnea-Hypopnea Index (AHI) index. Is it high enough for some applicaitons?
Can the author provide a table to compare the PBS method with other kinds of sleep monitoring methods? i.e., for using the same pressure bed sensor, what's the advantage of this paper's signal processing method? If using other kinds of sensor (such as ACC), what the advantage of this paper's method?
Round 2
Reviewer 1 Report
Figure and figure-text contents need to be improved for better readability.
Author Response
All the authors would like to thank the Reviewer for the suggestions provided. We have now modified the figures and zoomed their figure-text contents for better readibility. We have also revised the whole manuscript to improve English language and style.
Reviewer 3 Report
Accept
Author Response
All the authors would like to thank the Reviewer for the suggestions provided. We have now revised the whole manuscript to improve English language and style. We have also now modified the figures and zoomed their figure-text contents for better readibility.